# Neuroinflammatory Signature of Post-Traumatic Confusional State: The Role of Cytokines in Moderate-to-Severe Traumatic Brain Injury

**DOI:** 10.3390/ijms26178593

**Published:** 2025-09-04

**Authors:** Federica Piancone, Francesca La Rosa, Ambra Hernis, Ivana Marventano, Pietro Arcuri, Marco Rabuffetti, Jorge Navarro, Marina Saresella, Mario Clerici, Angela Comanducci

**Affiliations:** 1IRCCS Fondazione Don Carlo Gnocchi ONLUS, 20148 Milan, Italy; fpiancone@dongnocchi.it (F.P.); ahernis@dongnocchi.it (A.H.); imarventano@dongnocchi.it (I.M.); parcuri@dongnocchi.it (P.A.); jnavarro@dongnocchi.it (J.N.); msaresella@dongnocchi.it (M.S.); mario.clerici@unimi.it (M.C.); acomanducci@dongnocchi.it (A.C.); 2Department of Pathophysiology and Transplantation, University of Milan, 20122 Milan, Italy

**Keywords:** traumatic brain injury, post-traumatic confusional state, post-traumatic amnesia, neuroinflammation, cytokines, neurotransmitters, neurotrophins, IL-10, sleep efficiency, actigraphy, neurorehabilitation

## Abstract

Traumatic brain injury (TBI), a leading cause of mortality and disability, recognizes a primary, immediate injury due to external forces, and a secondary phase that includes inflammation that can lead to complications such as the post-traumatic confusional state (PTCS), potentially impacting long-term neurological recovery. An earlier identification of these complications, including PTCS, upon admission to intensive rehabilitation units (IRU) could possibly allow the design of personalized rehabilitation protocols in the immediate post-acute phase of moderate-to-severe TBI. The present study aims to identify potential biomarkers to distinguish between TBI patients with and without PTCS. We analyzed cellular and molecular mechanisms involved in neuroinflammation (IL-6, IL-1β, IL-10 cytokines), neuroendocrine function (norepinephrine, NE, epinephrine, E, dopamine), and neurogenesis (glial cell line-derived neurotrophic factor, GDNF, insuline-like growth factor 1, IGF-1, nerve growth factor, NGF, brain-derived growth factor, BDNF) using enzyme-linked immunosorbent assay (ELISA), comparing results between 29 TBI patients (17 with PTCS and 12 non-confused) and 34 healthy controls (HC), and correlating results with an actigraphy-derived sleep efficiency parameter. In TBI patients compared to HC, serum concentration of (1) pro-inflammatory IL-1β cytokine was significantly increased while that of anti-inflammatory IL-10 cytokine was significantly decreased; (2) NE, E and DA were significantly increased; (3) GDNF, NGF and IGF-1 were significantly increased while that of BDNF was significantly decreased. Importantly, IL-10 serum concentration was significantly lower in PTCS than in non-confused patients, correlating positively with an improved actigraphy-derived sleep efficiency parameter. An anti-inflammatory environment may be associated with better prognosis after TBI.

## 1. Introduction

Traumatic brain injury (TBI), a brain injury that is caused by an outside force (such as falls or vehicle accidents) is one of the leading causes of mortality and disability worldwide, often resulting in severe and long-lasting consequences that include disturbances of consciousness ranging from coma to post-traumatic confusional state (PTCS). PTCS is characterized by fundamental neurobehavioral traits that include disturbances to attention, disorientation, disturbances to memory. In addition, PTCS subjects manifest agitation, sleep–wake cycle disturbances, delusions, and confabulation [1].

Following the primary injury, caused by mechanical external forces, damage-associated molecular patterns (DAMPs) are released from damaged cells and initiate a complex cascade of mechanisms that lead to post-traumatic neuroinflammation [2]. DAMPs bind to patterns recognition receptors (PRRs) on immune cells residing in the Central Nervous System (CNS), triggering the production of cytokines, chemokines, and other immune modulators, that recruit immune cells from the periphery [3].

The ensuing neuroinflammatory response can be beneficial and promote recovery if limited in time [4]; however, when excessive or prolonged, it is associated with brain ageing, neurodegeneration, and dementia [5]. Among pro-inflammatory mediators, IL-1β has been reported to be increased in TBI [6]. Notably, IL-1β neutralization was shown to reduce microglial activation [6], cerebral edema, and neurodegeneration in TBI animal models [7,8].

Additionally, IL-6 production is also significantly elevated after TBI; together with IL-1β, IL-6 stimulates the release of catecholamines [9,10] which, in turn, perpetuate the pro-inflammatory state and contribute to worse outcomes [11,12]. TBI-activated microglia have also been shown to be detrimental for neurogenesis [13], and the production of pro-inflammatory cytokines by M1-like brain injury-activated microglia suppress neurogenesis [14,15]. Additionally, intracortical or intraperitoneal administration of lipopolysaccaride (LPS), an agent able to induce a strong immune response, significantly impairs neurogenesis by reducing BDNF gene expression [16], decreasing the survival of newly generated neurons and limiting the differentiation of new cells into neurons [17].

In vitro studies on TBI patients showed that neuronal/stem progenitor cells (NSCs) protein markers, including neuronal migration protein doublecortin (DCX), TUC4, polysialic acid–neural cell adhesion molecule (PSA-NCAM), SOX2 and NeuroD, and Ki67 proliferative marker, are increased, suggesting that TBI induces neurogenesis in human brain [18]. Insulin-like growth factor 1 (IGF-1), a neuroendocrine regulator of the CNS, was upregulated in animal models of TBI, likely supporting neural repair [19]. Likewise, exogenous administration of nerve growth factor (NGF), another neurotrophin (NT), was shown to reduce brain cell death [20].

Notably, neurogenesis and gliogenesis in the developing nervous system are also regulated by extrinsic factors, including physical activity and sleep. In this context, it is important to underline that the great majority of TBI patients experience disturbances in sleep–wake cycle regulation [21] which may contribute to delayed recovery [22]. Indeed, sleep is fundamentally important for memory consolidation, clearance of brain debris, reduction in neuroinflammation, and enhancement of neurogenesis and neuroplasticity—processes that are critical for recovery from TBI.

Most patients who survive moderate-to-severe TBI experience a phase of recovery characterized by a post-traumatic confusional state (PTCS), during which symptoms such as agitation, irritability, disorientation, perceptual impairments, sleep disruption, and reduced arousal may complicate clinical management and lead to poorer functional outcomes post-rehabilitation [21].

Although PTCS represents a clinically relevant phase in TBI recovery, its biological correlates remain largely unexplored. Our study aims to identify peripheral biomarkers related to inflammation, neuroendocrine function, and neurotrophic signalling that differentiate TBI patients with and without PTCS in the early post-acute phase by investigating their potential association with sleep disturbances.

## 2. Results

### 2.1. Demographic and Clinical Characteristics of Study Subjects

Demographic and clinical characteristics of all participants—including Disability Rating Scale (DRS) scores at admission and time from injury to recruitment (in days) for patients—are shown in Table 1. Results showed a significant difference in DRS scores between patients with and without PTCS as expected.

### 2.2. Pro- and Anti-Inflammatory Cytokines in TBI Patients with and Without PTCS and in Healthy Controls

Results showed that IL-1 β serum concentration was significantly increased in TBI subjects (median = 41 pg/mL) compared to HC (median = 3.5 pg/mL) (*p* < 0.0001) (Figure 1A). Augmented IL-1β serum concentration was maintained when TBI was divided up in PTCS and non-confused patients (vs. HC: median PTCS = 39.9 pg/mL: *p* < 0.05; median non-confused = 42.8 pg/mL; *p* < 0.0001) suggesting the presence of inflammation in traumatic brain subjects (Figure 1B).

IL-6 serum concentration was significantly increased as well in TBI subjects compared to HC (median = 4.4 pg/mL and 2.3 pg/mL, respectively; *p* < 0.005) (Figure 1C). This difference was maintained when in comparison between non-confused subjects and HC (*p* < 0.05) (Figure 1D).

We next evaluated serum concentration of the anti-inflammatory IL-10. Results showed a significant decrease in IL-10 serum concentration in TBI patients compared to HC (*p* < 0.005) (Figure 2A). Notably, IL-10 serum concentration could differentiate between subgroups of TBI patients as it was greatly reduced in PTCS compared to non-confused patients (*p* ≤ 0.0005) (Figure 2B).

### 2.3. Neurotransmitters in TBI Patients with and Without PTCS and in Healthy Controls

Elevated catecholamine levels might be suggested to be associated with a worse clinical outcome in TBI. We evaluated concentrations of norepinephrine (NE), epinephrin (E), and dopamine (DA) in all individuals recruited for the study. Results showed that serum concentration of all catecholamines was increased in TBI patients compared to HC (*p* < 0.0001 for all the comparisons) (Figure 3A–C). No differences, though, could be detected when PTCS and non-confused TBI patients were compared (Figure 3D–F).

### 2.4. Neurotrophins of in TBI Patients with and Without PTCS and in Healthy Controls

Neurotrophins are involved in plasticity and can attenuate neuronal injury. To investigate regeneration and neurogenesis in TBI, we evaluated serum concentration of glial cell line-derived neurotrophic factor (GDNF), nerve growth factor (NGF), insulin-like growth factor-1 (IGF-1), and brain-derived neurotrophic factor (BDNF). Results showed that GDNF, NGF, and IGF-1 were significantly increased in serum of TBI subjects compared to HC (Figure 4A–C) (*p* < 0.005; *p* < 0.0005; *p* < 0.05, respectively), while BDNF concentration was higher in HC subjects compared to TBI (*p* < 0.0001) (Figure 4D).

The analysis of PTCS and non-confused TBI subgroups compared to HC confirmed the same pattern observed in the overall TBI group. Specifically, GDNF levels were significantly increased in both PTCS (*p* < 0.05) and non-confused patients (*p* < 0.05) compared to HC (Figure 5A); NGF levels were also significantly higher in PTCS (*p* < 0.005) and non-confused TBI patients (*p* = 0.05) compared to HC (Figure 5B); and IGF-1 levels were significantly elevated in the non-confused group compared to HC (*p* < 0.05) (Figure 5C). Conversely, BDNF levels were significantly decreased in both PTCS (*p* < 0.001) and non-confused TBI patients (*p* < 0.005) compared to HC (Figure 5D).

### 2.5. Sleep–Wake Disturbances and Cytokine Profiles

Sleep–wake disturbances are increasingly recognized as a serious consequence and a barrier to recovery, and affect up to 70% of individuals with TBI.

Building on previous findings by Makley et al. [23], which suggest differences in sleep efficiency between PTCS and non-confused subjects following moderate to severe TBI, we utilized actigraphy to analyze sleep–wake parameters in a similar cohort. Our results confirmed a significant decrease in sleep efficiency (*p* < 0.05) in patients with PTCS (median = 73%) compared to those in the non-confused group (median = 85%) (Figure 6A). Additionally, we explored possible correlations between sleep efficiency and serum concentrations of anti-inflammatory and pro-inflammatory cytokines, neuroendocrine factors, and neurotrophins. Our findings revealed a significant positive correlation between serum IL-10 concentration and sleep efficiency scores (*p* < 0.05, Rsp = 0.43) (Figure 6B), suggesting that higher levels of the anti-inflammatory cytokine IL-10 are associated with improved sleep quality, which may contribute to more favourable conditions for rehabilitation.

## 3. Discussion

Traumatic brain injury is a leading cause of mortality and disability worldwide [24]. The primary injury, caused by external mechanical forces, occurs immediately and it is typically unpredictable. In contrast, the second phase of injury is delayed in time and involves a cascade of biochemical and cellular events, such as inflammation, ischemia, and oxidative stress, that exacerbates the initial damages. This phase can precipitate various complications, including increased intracranial pressure and the emergence of a PTCS, which profoundly impact rehabilitation outcomes and long-term recovery.

The extent of the secondary injury depends upon the binding of damage-associated molecular patterns (DAMPs), released from damaged and/or dying cells, to Pattern Recognition Receptors (PRRs) on myeloid and dendritic cells, initiating a neuroinflammatory cascade [25,26]. This process can last for months or years and greatly increases the risk of developing neurodegenerative diseases [27].

Mouse models of TBI have shown that microglia activation persists up to 30 days after injury [28], while human studies have shown that microglia activation can be observed even years after injury [29].

Inflammation is not limited to the SNC but extends to the periphery. In our cohort, peripheral inflammation was reflected by significantly elevated serum levels of IL-1β and IL-6 in the peripheral blood of TBI patients. This result is in line with results from other studies that showed high IL-1β serum concentration in TBI subjects and suggested this to be associated with a higher risk of suffering from cognitive impairment [30]. TBI-associated neuroinflammation has also been associated with sympathetic nervous system (SNS) activation, resulting in massive release of catecholamines [31]. In line with this, our results revealed significantly higher levels of epinephrin, norepinephrine, and dopamine in TBI patients, in conjunction with increased IL-1β and IL-6, indicating a sustained pro-inflammatory and neuroendocrine response.

The vast majority of patients with moderate-to-severe TBI experience a PTCS during recovery—a phase historically referred to as post-traumatic amnesia (PTA) but now recognized as a distinct clinical entity encompassing a broader spectrum of symptoms, including disorientation, agitation, attention deficits, and disturbances of the sleep–wake cycle [21]. These symptoms interfere with a patient’s ability to engage in rehabilitation, thereby complicating the recovery process. Notably, the duration of PTCS is one of the most robust prognostic indicators in TBI, with longer durations associated with poorer functional outcomes [32].

To better understand the biological correlates of PTCS, we profiled serum levels of neuroinflammatory, neuroendocrine, and neurotrophic markers in TBI patients with and without PTCS.

Neuroplasticity, a key component of CNS recovery, is supported by the release of neurotrophic factors. Our findings revealed significantly increased serum levels of GDNF, NGF, IGF-1, and BDNF serum in TBI subjects. GDNF, in particular, is known for promoting the survival of catecholaminergic neurons [33]. Indeed, GDNF is able to almost totally rescue the number of these neurons, both in tissue culture and in various lesion models in vivo [34]. Our results indicate that the GDNF serum concentration is significantly increased in TBI patients, with higher levels observed in non-confused individuals. This suggests that the remodelling of catecholamine neurons occurs after trauma, and is more efficient in non-confused patients.

NGF, the first discovered neurotrophic factor, was also increased in TBI patients. This result aligns with the observed increase in IL-1β levels in TBI, as Il-1β elevation initiates a cascade of events leading to upregulation of NGF in brain tissues [35]. On the contrary, induction of IL-1β levels has been shown to robustly decrease BDNF mRNA in the hippocampus [36] and BDNF is known to trigger protein synthesis in a late phase of the synaptic consolidation [37].

Our study confirmed that, unlike GDNF and NGF, which appear to support neuroprotective efforts, BDNF serum levels were decreased in both PTCS and non-confused TBI patients, compared to HC.

This suggests that while GDNF and NGF may contribute to a neuroprotective response in TBI, BDNF does not support this response in the same way.

Sleep has emerged as a key modulator of neuroplasticity and inflammation in TBI recovery [38]. It is essential for memory consolidation, elimination of brain debris, reduction in neuroinflammation, and enhancement of neuroplasticity. Patients with sleep disturbances tend to have delayed recovery times [39] and show higher levels of systemic inflammation. This aligns with the observed significative increase in serum IL-10 production among non-confused subjects compared to their confused counterparts, despite the overall decreased levels of anti-inflammatory cytokines in TBI compared to healthy controls. Our results also revealed that IL-10 positively correlated with sleep efficiency (*p* = 0.04, Rsp = 0.43), supporting the hypothesis that sleep plays an anti-inflammatory role in the post-traumatic brain.

In summary, our findings provide novel insights into the neurobiological underpinnings of PTCS and its association with systemic inflammation, catecholamine release, neurotrophic activity, and sleep. These results deepen our understanding of inflammation-driven complications in TBI and may contribute to the development of targeted therapeutic strategies aimed at improving neuro-recovery and rehabilitation outcomes.

Limitations: this study has some limitations that must be acknowledged. First, the small sample size (29 patients with TBI (17 with PTCS and 12 without) and 34 controls) limits the statistical power of the analysis. Due to the limited number of subjects, the correction for multiple comparisons was not applied as it results in a reduction in statistical significance in few tests. Further, this study showed a sex imbalance distribution; finally, this study does not provide longitudinal follow-up to assess clinical outcomes or functional recovery. However, a rehabilitation programme on TBI patients is ongoing and data will be provided in a new work.

## 4. Materials and Methods

### 4.1. Patients Enrolled in the Study

Twenty-nine subjects with a diagnosis of subacute TBI were consecutively recruited at the Intensive Rehabilitation Unit (IRU) for Acquired Brain Injury of IRCCS Fondazione Don Carlo Gnocchi ONLUS in Milan, Italy. The medical team of the IRU has long been engaged in the epidemiological study of neurological disorders and, with patients’ informed consent, actively participates in major therapeutic trials for the development of new pharmacological treatments, in collaboration with national and international. The multidisciplinary rehabilitation programme, requiring a high level of care, is aimed at restoring the highest possible degree of patient autonomy and ensuring the best conditions for social and occupational reintegration, and includes a combination of cognitive therapy, physical therapy, occupational therapy, and speech therapy.

Thirty-four age-and-sex-matched healthy controls (HC) were enrolled as well.

To ensure clinical homogeneity, inclusion criteria were as follows: (1) subacute TBI (within three months post-injury); (2) moderate to severe injury severity; and (3) presence of diffuse axonal injury confirmed by neuroimaging. TBI severity was determined according to established clinical criteria [24], including the Glasgow Coma Scale (GCS) score at admission, duration of loss of consciousness (LOC), and post-traumatic amnesia (PTA). Patients were classified as moderate-to-severe when they had a GCS ≤ 12 and/or prolonged LOC (>30 min) or PTA (>24 h).

Exclusion criteria included the following: (1) age < 18 or >75 years; (2) a previous neurological or psychiatric disorder (particularly neurodegenerative or acquired conditions affecting cognitive domains, e.g., dementia); (3) medical instability; (4) the presence of a disorder of consciousness at IRU admission; (5) use of nonsteroidal anti-inflammatory drugs (NSAIDs) and corticosteroids.

Healthy controls were recruited among familiars, visiting patients, and healthy subjects recruited among the IRCCS Fondazione Don Carlo Gnocchi users for blood tests. Inclusion criteria: (1) candidates who agreed to participate in this study and signed a written consent form; (2) subjects are healthy males and females aged 18–75. Exclusion criteria were as follows: (1) nonsteroidal anti-inflammatory drugs (NSAIDs) and corticosteroids; (2) autoimmune or inflammatory related pathologies (cancer, diabetes, infections); (3) neurological or psychiatric disorder.

### 4.2. Ethics Approval

The study complies with the ethical principles of the Declaration of Helsinki; written informed consent was provided by all the individuals involved in the study and was designed according to a protocol approved by the local ethics committee of IRCCS Fondazione Don Carlo Gnocchi (Protocol number #02_20/05/2021).

### 4.3. Clinical Evaluation of Post-Traumatic Confusional State

Upon admission to the IRU, patients with TBI were evaluated to diagnose PTCS using the Confusion Assessment Protocol (CAP), a multidimensional scale recommended for PTCS diagnosis that includes the Galveston Orientation and Amnesia Test (GOAT) as one of its core items [22]. The GOAT assesses orientation to person, place, time, and event, with scores ranging from 0 to 100; a score below 75 is indicative of impaired orientation and amnesia, consistent with PTCS [40]. The CAP measures seven key multidomain symptoms of PTCS: cognitive impairment, disorientation (as assessed by the GOAT), agitation, symptom fluctuations, nighttime sleep disturbances, decreased daytime arousal, and psychotic symptoms [22]. Each domain contributes one point to a total CAP score that ranges from 0 (no confusional symptoms) to 7 (all symptoms present). A diagnosis of PTCS is made when at least four symptoms are present, or three if one of them is disorientation.

To establish a dichotomous classification of “confused” vs. “not confused”, we adopted the following criteria: patients were classified as confused (PTCS present) if they met the CAP diagnostic threshold for PTCS (i.e., ≥4 symptoms, or 3 including disorientation). Patients were classified as not confused (PTCS absent) only if they had at least two consecutive daily assessments that were negative on the CAP.

### 4.4. Actigraphic Evaluation of Sleep

Actigraphs are small and non-invasive wristwatch accelerometers that monitor rest and activity cycles, by recording movement data, to assess sleep-related parameters such as total sleep time, sleep efficiency (SE), sleep latency, wake after sleep onset, and number of awakenings, providing insights into the sleep–wake cycle and circadian rhythms [40].

For this study, all subjects were equipped with a wearable actigraph (Geneactiv, Activinsights, Huntingdon, UK) on the wrist of their non-paralyzed arm, ensuring the arm was free from fractures or open wounds. The device was worn for seven consecutive days during their hospitalization in the IRU and the blood sampling was performed during the same week. The wrist-worn tri-axial sensor of the actigraph continuously recorded acceleration at a sampling rate of 100 Hz throughout the entire seven-day period. The raw data collected were subsequently downloaded and analyzed using ad hoc software implemented in Matlab (v. R2017b, The Mathworks, Natick, MA, USA). Analysis involved segmenting the acceleration vector norm profile into one-minute epochs, yielding a comprehensive view of the patient’s activity levels across each day. Sleep Efficiency (SE) was calculated as the percentage of time spent asleep (i.e., time without noticeable movement) during the designated bedtime period (typically from 11 PM to 7 AM), effectively quantifying the actual sleep a patient manages to achieve during the standard sleep period defined by the IRU.

### 4.5. Serum Sample Collection

Blood was drawn from all subjects enrolled in the study and processed to obtain serum for analysis. Serum was obtained from blood by centrifugation (2000× *g* × 10′ at room temperature) and stored within 1 h and 30 min from the collection in several aliquots at −80 °C immediately after sampling in order to avoid freeze–thaw cycles and was thawed upon performing experiments.

### 4.6. ELISA

#### 4.6.1. Cytokines

IL-1β, IL-6, and IL-10 serum concentrations was determined by sandwich immunoassays according to the manufacturer’s recommendations (IL-1β, IL-6 and IL10, Invitrogen, Thermofisher, Carlsbad, CA, USA); catalogue numbers n° KHC0011, n° BMS214 and n° KHC0101, respectively; IL-6, and sgp130, Biotechne, Minneapolis, MN, USA; catalogue numbers n° SPCKB-PS-009384 and n° DGP00, respectively). Samples were read on a plate reader (Sunrise, Tecan, Mannedorf, Switzerland) and optical density (OD) was determined at 450/620 nm. The measured absorbance is proportional to the concentration of cytokines in serum. Sensitivity (S) and assay range (AR) were as follows: S: IL-1β = 1 pg/mL; IL-6 = 0.1 pg/mL; IL-10 = 1 pg/mL. AR: IL-1β = 3.9–250 pg/mL; IL-6 = 0.28–2652 pg/mL; IL-10 = 7.8–500 pg/mL. Measured absorbance is proportional to IL-1β, IL-6, and IL-10 concentration in serum expressed in pg/mL and calculated by dividing the optical density (OD) measurement generated from the assay by OD cut-off calibrator. All the experiments were performed in duplicate.

#### 4.6.2. Neurotransmitters

Quantification of the neurotransmitters norepinephrine (Abcam, Cambridge, UK; catalogue number n° ab287789), epinephrine (Abcam, catalogue number n° ab287778), and dopamine (Abcam, catalogue number n° ab285238) was performed in sera by sandwich immunoassay according to the manufacturer’s recommendations. Sensitivity (S) and assay range (AR) were as follows: S: norepinephrine = 9.375 pg/mL; epinephrine = 4.68 pg/mL; dopamine = 0.938 ng/mL; AR: norepinephrine = 15.625–1000 pg/mL; epinephrine = 7.8–500 pg/mL; dopamine = 1.56–100 ng/mL. As indicated above, measured absorbance is proportional to serum concentration of neurotransmitters expressed in pg/mL or ng/mL and was calculated by dividing the optical density (OD) measurement generated from the assay by OD cut-off calibrator. All the experiments were performed in duplicate.

#### 4.6.3. Neurotrophins

Serum concentration of the neurotrophins GDNF (Abcam, catalogue number n° ab100525), NGF (Abcam, catalogue number n° ab193760), BDNF (R&D, Biotechne, Minneapolis, MN, USA; catalogue number n° DBNT00), and IGF-1 (Abcam, catalogue number n° ab 211651) was measured by sandwich immunoassays according to the manufacturer’s recommendations. Sensitivity (S) and assay range (AR) were as follows: S: GDNF = 4 pg/mL; NGF = 6 pg/mL; BDNF= 1.35 pg/mL; IGF-1 = 11.97 pg/mL; AR: GDNF = 2.74–2000 pg/mL; NGF = 27.3–1750 pg/mL; BDNF = 15.6–1000 pg/mL; IGF-1 = 93.75–6000 pg/mL. Also in this case, measured absorbance is proportional to serum concentration of neurotransmitters expressed in pg/mL or ng/mL and calculated by dividing the optical density (OD) measurement generated from the assay by OD cut-off calibrator. All the experiments were performed in duplicate.

### 4.7. Statistical Analysis

Not-normally distributed data were expressed as median and interquartile range (IQR), and comparison were analyzed by Mann–Whitney rank test. Correlations were analyzed by Spearman’s rank correlation coefficient (Rsp). Data analysis was performed using the MedCalc statistical package (v. 11.5.0.0, MedCalc Software bvba, Mariakerke, Belgium). *p*-values of less than 0.05 were considered statistically significant.

## Figures and Tables

**Figure 1 ijms-26-08593-f001:**
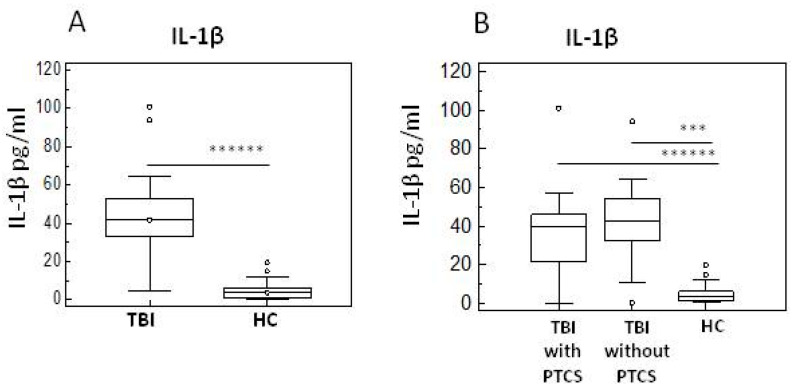
Pro-inflammatory cytokines in TBI Patients with and without PTCS and in healthy controls. Results of (IL)-1β and IL-6 serum concentration in TBI and HC subjects (Panel (**A**,**C**)) and in TBI patients with and without PTCS (**B**,**D**) were summarized in box-and-whiskers plots (the boxes stretch from the 25 to the 75 percentiles; the line across the boxes indicates the median values; the lines stretching from the boxes indicate extreme values; open circles represent outliers. Statistical significance is shown * *p* < 0.05; *** *p* < 0.005; ****** *p* < 0.0001).

**Figure 2 ijms-26-08593-f002:**
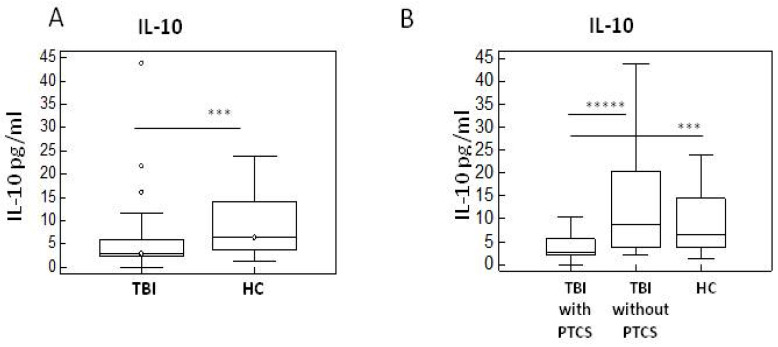
Anti-inflammatory IL-10 in TBI Patients with and without PTCS and in healthy controls. Results of IL-10 in TBI and HC subjects (**A**) and in TBI with and without PTCS (**B**) were summarized in box-and-whiskers plots (the boxes stretch from the 25 to the 75 percentiles; the line across the boxes indicates the median values; the lines stretching from the boxes indicate extreme values; statistical significance is shown; open circles represent outliers. *** *p* < 0.005; ***** *p* < 0.0005).

**Figure 3 ijms-26-08593-f003:**
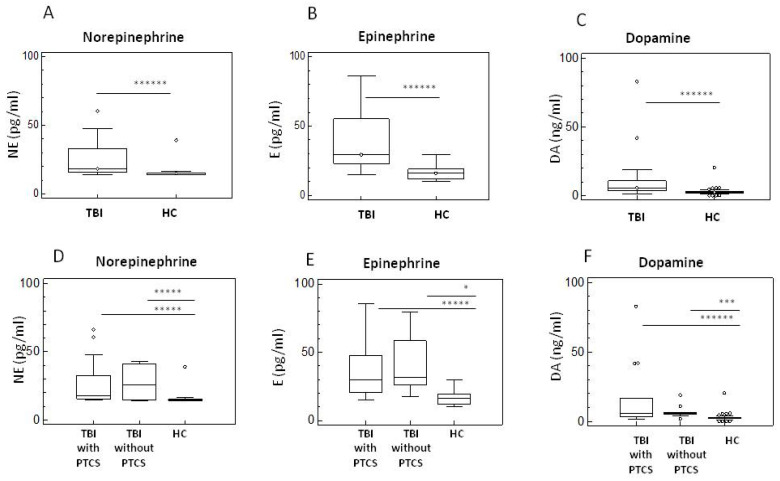
Neurotransmitters levels in serum of TBI Patients with and without PTCS and in HC. Summary results of comparison between TBI and HC were presented in panels (**A**) (Epinephrine), (**B**) (Norepinephrine), and (**C**) (Dopamine). Comparison between TBI with and without PTCS were summarized in panels (**D**) (Norepinephrine), (**E**) (Epinephrine), and (**F**) (Dopamine). Data are presented as box plots; the boxes stretch from the 25 to the 75 percentiles; the line across the boxes indicates the median values; the lines stretching from the boxes indicate extreme values; open circles represent outliers. Statistical significance is shown. * *p* < 0.05; *** *p* < 0.005; ***** *p* < 0.0005; ****** *p* < 0.0001.

**Figure 4 ijms-26-08593-f004:**
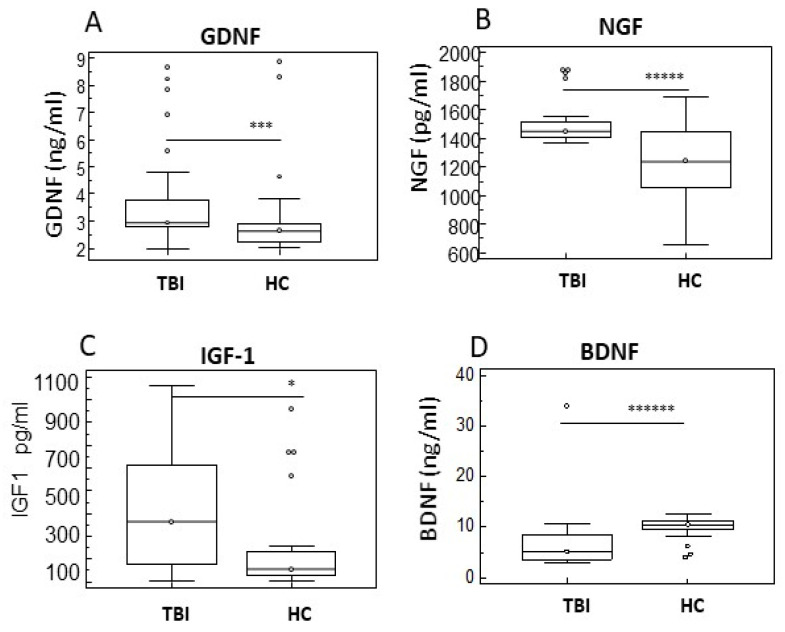
Neurotrophins in serum of TBI Patients with and without PTCS and in healthy controls. Box-plots representing comparison between serum levels of GDNF (**A**), NGF (**B**), IGF-1 (**C**), and BDNF (**D**) in TBI subjects compared to HC are shown. Data are presented as box plots; the boxes stretch from the 25 to the 75 percentiles; the line across the boxes indicates the median values; the lines stretching from the boxes indicate extreme values; open circles represent outliers. Statistical significance is shown. * *p* < 0.05; *** *p* < 0.005; ***** *p* < 0.0005; ****** *p* < 0.0001.

**Figure 5 ijms-26-08593-f005:**
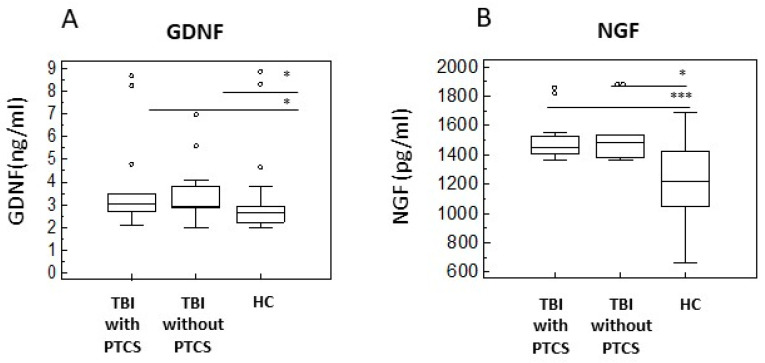
Neurotrophins in serum of TBI patients with and without PTCS and in healthy controls. Box-plots represent comparison between serum levels of GDNF (**A**), NGF (**B**), IGF-1 (**C**), and BDNF (**D**) in TBI subjects with and without PTCS; the boxes stretch from the 25 to the 75 percentiles; the line across the boxes indicates the median values; the lines stretching from the boxes indicate extreme values; open circles represent outliers. Statistical significance is shown. * *p* < 0.05; *** *p* < 0.005; **** *p* < 0.001.

**Figure 6 ijms-26-08593-f006:**
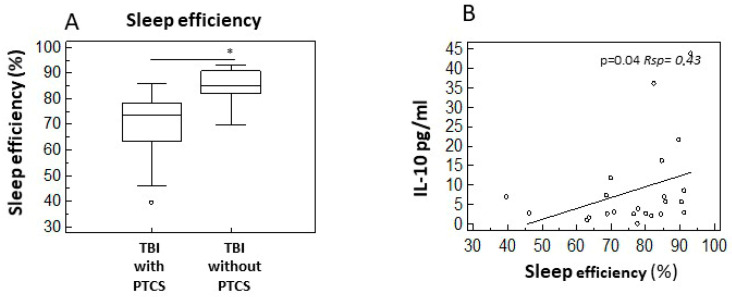
Sleep–wake disturbances and cytokine profile in TBI subjects. Results of the comparison of sleep efficiency between TBI subjects with and without PTCS: (**A**) A scatter plot with regression line indicating the correlation between sleep efficiency and IL-10 is displayed in panel (**B**). Circles in panel (**B**) represent each data sample. Statistical significance is shown. * *p* < 0.05. Rsp: Spearman rank correlation coefficient.

**Table 1 ijms-26-08593-t001:** Demographic and clinical characteristics of the study populations.

Variable	Patients with PTCS	Patients Without PTCS	Healthy Controls	*p*-Value
Age, years	42.6 ± 19.5	48.2 ± 15.6	52.5 ± 1.7	-
Gender (M:F)	16:1	9:3	16:18	-
Days from injury	35.4 ± 5.5	33.5 ± 3.5	-	<0.05
Disability rating scale at admission	17 (15–19)	13 (8–16)	-	n.s.

Quantitative data are expressed as median and interquartile range (DRS) mean ± standard deviation (Age, Days from injury). M: Male; F: Female; PTCS: Post-traumatic confusional state; n.s.: not statistically significant.

## Data Availability

Data generated during the current study are available from the corresponding author on reasonable request.

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
