# Peer review of "Neuroinflammatory Signature of Post-Traumatic Confusional State: The Role of Cytokines in Moderate-to-Severe Traumatic Brain Injury"

_ijms, 2025, doi:10.3390/ijms26178593_

Round 1

Reviewer 1 Report

Comments and Suggestions for Authors

The submitted manuscript presents an interesting and clinically relevant study exploring biomarkers associated with post-traumatic confusional state (PTCS) in patients with moderate-to-severe traumatic brain injury (TBI). The study addresses an important unmet need in neurorehabilitation: the early identification of neurobiological markers that could support personalized therapeutic strategies in the immediate post-acute phase of TBI. The aim to differentiate TBI patients with and without PTCS based on peripheral biomarkers is timely and scientifically valuable.

Below are my main comments and suggestions for improvement:

1. Introduction

- Several abbreviations (e.g., PTCS, TBI, IRU) appear in the Introduction and throughout the manuscript without being clearly defined at first use. These should be consistently introduced and explained.

- The stated aim — to improve rehabilitation protocols by investigating the complex relationship between neurobiological processes, PTCS, and sleep disturbances — appears too ambitious and not fully aligned with the presented findings. The data do not directly evaluate rehabilitation outcomes or the impact of any therapeutic protocols. I recommend reformulating the objective in a way that more accurately reflects the scope and findings of the study, for example: “This study aims to identify peripheral biomarkers related to inflammation, neuroendocrine function, and neurotrophic signaling that differentiate TBI patients with and without PTCS in the early post-acute phase.”

2. Results

- The results are presented clearly and are generally well-structured.

- The formatting of p-values should be revised for consistency. The correct format is p < ... (not p = ... when the value is not exact).

- Please clarify whether the difference in sex distribution among groups was statistically tested. The observed distribution — PTCS (M:F = 16:1), non-PTCS (M:F = 9:3), and healthy controls (M:F = 16:18) — suggests potential imbalance. If not analyzed, this should be addressed, and the results reported, as it may represent a confounding factor.

3. Discussion

- The discussion would benefit from a dedicated paragraph addressing the study’s limitations. Most notably: the relatively small sample size; the imbalance in sex distribution across study groups; the lack of longitudinal follow-up to assess clinical outcomes or functional recovery; potential effects of unmeasured confounders (e.g., comorbidities, medications).

4. Materials and Methods

- please include information on how healthy controls were recruited and what inclusion/exclusion criteria were applied.

- since inflammatory markers were analyzed, it is critical to specify whether participants with inflammatory-related conditions (e.g., infections, cancer, autoimmune diseases) were excluded. If not, this introduces significant bias.

- clearly define the inclusion/exclusion criteria for the control group. Were they matched for age/sex or screened for neurological/psychiatric/ inflammatory (cancer, infection, autoimmune diseases, etc.) disorders?

- elaborate on how TBI severity was determined (e.g., Glasgow Coma Scale, imaging findings, duration of unconsciousness) and how patients were classified into the “moderate-to-severe” group.

- Regarding cytokine analysis:

Provide more detailed information on sample collection and storage procedures, including handling of serum samples, centrifugation protocols, and storage duration.

The phrase “serum was thawed upon performing experiments” raises concern about possible multiple freeze–thaw cycles, which may affect cytokine stability. Please clarify whether samples were aliquoted and frozen only once.

Time of sample storage is important, particularly for short half-life cytokines like IL-10. Please report the average or range of storage duration.

Note that blood is collected from participants, not serum. The correct terminology is that blood was drawn and processed to obtain serum for analysis.

- The statistical analysis section should be expanded:

Specify which tests were used to assess normality (e.g., Shapiro-Wilk, Kolmogorov-Smirnov).

Indicate how normally and non-normally distributed variables were analyzed.

Clarify how correlation coefficients (e.g., Spearman’s rho) were interpreted in terms of strength and significance.

Conclusion

The study offers novel insights into peripheral biomarkers associated with PTCS following TBI, with potential implications for early diagnosis and monitoring. However, certain methodological details and clarifications are essential to enhance transparency and scientific rigor. I recommend major revisions before the manuscript can be considered for publication.

Author Response

Reviewer 1

The submitted manuscript presents an interesting and clinically relevant study exploring biomarkers associated with post-traumatic confusional state (PTCS) in patients with moderate-to-severe traumatic brain injury (TBI). The study addresses an important unmet need in neurorehabilitation: the early identification of neurobiological markers that could support personalized therapeutic strategies in the immediate post-acute phase of TBI. The aim to differentiate TBI patients with and without PTCS based on peripheral biomarkers is timely and scientifically valuable.

Below are my main comments and suggestions for improvement:

  1. Introduction
  • Several abbreviations (e.g., PTCS, TBI, IRU) appear in the Introduction and throughout the manuscript without being clearly defined at first use. These should be consistently introduced and explained.
    Response: Thank you for your kind suggestion. The abbreviations have been clarified in the text as follows:

    Line 42. Traumatic brain injury (TBI), a brain injury that is caused by an outside force (such as falls or vehicle accidents),

    Line 46. PTCS is characterized by fundamental neurobehavioral traits that include disturbances of attention, disorientation, disturbances of memory. In addition, PTCS subjects manifest agitation, sleep-wake cycle disturbances, delusions and confabulation [1].

    Line 383. The medical team of the IRU has long been engaged in the epidemiological study of neurological disorders and, with patients’ informed consent, actively participates in major therapeutic trials for the development of new pharmacological treatments, in collaboration with national and international. The multidisciplinary rehabilitation program, requiring a high level of care, is aimed at restoring the highest possible degree of patient autonomy and ensuring the best conditions for social and occupational reintegration and include a combination of cognitive therapy, physical therapy, occupational therapy, and speech therapy.

  • The stated aim — to improve rehabilitation protocols by investigating the complex relationship between neurobiological processes, PTCS, and sleep disturbances — appears too ambitious and not fully aligned with the presented findings. The data do not directly evaluate rehabilitation outcomes or the impact of any therapeutic protocols. I recommend reformulating the objective in a way that more accurately reflects the scope and findings of the study, for example: “This study aims to identify peripheral biomarkers related to inflammation, neuroendocrine function, and neurotrophic signaling that differentiate TBI patients with and without PTCS in the early post-acute phase.”
    Response: Thank you for your comment. We reformulated the objective of the study as follows:

    Line 92 Our study aims to identify peripheral biomarkers related to inflammation, neuroendocrine function, and neurotrophic signaling that differentiate TBI patients with and without PTCS in the early post-acute phase by investigating their potential association with sleep disturbances.
  1. Results
  • The results are presented clearly and are generally well-structured.
  • The formatting of p-values should be revised for consistency. The correct format is p < ... (not p = ... when the value is not exact).
    Response: Thank you. We modified the p-values according to you suggestion in the text and in the figures.

  • Please clarify whether the difference in sex distribution among groups was statistically tested. The observed distribution — PTCS (M:F = 16:1), non-PTCS (M:F = 9:3), and healthy controls (M:F = 16:18) — suggests potential imbalance. If not analyzed, this should be addressed, and the results reported, as it may represent a confounding factor.
    Response: t is worth noting that the male predominance in the PTCS and non-PTCS groups reflects the well-documented epidemiology of TBI which is consistently more frequent in men due to higher exposure to risk factors such as traffic accidents and occupational injuries (Eom et al). Nevertheless, we acknowledge that sex could represent a confounding factor, and this is now addressed in the limitations section.

    Eom KS, Kim JH, Yoon SH, Lee SJ, Park KJ, Ha SK, Choi JG, Jo KW, Kim J, Kang SH, Kim JH. Gender differences in adult traumatic brain injury according to the Glasgow coma scale: A multicenter descriptive study. Chin J Traumatol. 2021 Nov;24(6):333-343. doi: 10.1016/j.cjtee.2021.06.004. Epub 2021 Jun 9. PMID: 34275712; PMCID: PMC8606602.
  1. Discussion
  • The discussion would benefit from a dedicated paragraph addressing the study’s limitations. Most notably: the relatively small sample size; the imbalance in sex distribution across study groups; the lack of longitudinal follow-up to assess clinical outcomes or functional recovery; potential effects of unmeasured confounders (e.g., comorbidities, medications).
    Response: Thank you for your kind suggestion. We added a paragraph showing the limitations of the study at the end of the discussion.

    Line 374 Limitations: this study has some limitations that must be acknowledged. First, the small sample size (29 patients with TBI (17 with PTCS and 12 without) and 34 controls) limits the statistical power of the analysis. Further, this study showed a sex imbalance distribution; finally, this study does not provide longitudinal follow-up to assess clinical outcomes or functional recovery. However, rehabilitation program on TBI patients is ongoing and data will be provided in a new work.
  1. Materials and Methods
  • please include information on how healthy controls were recruited and what inclusion/exclusion criteria were applied.
    Response: Thank you. Information of HC have been included in paragraph 4.1. Patients Enrolled in the Study as follows:

    Line 401. Healthy controls were recruited among familiars, visiting patients and healthy subjects recruited among the IRCCS Fondazione Don Carlo Gnocchi users for blood tests. Inclusion criteria: (1) candidates who agreed to participate in this study and signed a written consent form; (2) subjects are healthy males and females aged 18-75. Exclusion criteria were: (1) nonsteroidal anti-inflammatory drugs (NSAIDs) and corticosteroids; (2) autoimmune or inflammatory related pathologies (cancer, diabetes, infections; (3) neurological or psychiatric disorder, including sleep-related disorders.

  • since inflammatory markers were analyzed, it is critical to specify whether participants with inflammatory-related conditions (e.g., infections, cancer, autoimmune diseases) were excluded. If not, this introduces significant bias.
    Response: Thank you for your comment. This aspect has been introduced in the exclusion criteria as follows:

    Line 399 Exclusion criteria included: (1) age <18 or >75 years; (2) a previous neurological or psychiatric disorder (particularly neurodegenerative or acquired conditions affecting cognitive domains, e.g., dementia); (3) autoimmune or inflammatory related pathologies (cancer, diabetes, infections); (4) medical instability; (5) the presence of a disorder of consciousness at IRU admission; (6) use of nonsteroidal anti-inflammatory drugs (NSAIDs) and corticosteroids; (7) previously diagnosed sleep disorders.

  • clearly define the inclusion/exclusion criteria for the control group. Were they matched for age/sex or screened for neurological/psychiatric/ inflammatory (cancer, infection, autoimmune diseases, etc.) disorders?
    Response: Thank you again. Inclusion/exclusion criteria for control group have been introduced in 4.1. Patients Enrolled in the Study paragraph as follows:

    Line 401. Healthy controls were recruited among familiars, visiting patients and healthy subjects recruited among the IRCCS Fondazione Don Carlo Gnocchi users for blood tests. Inclusion criteria: (1) candidates who agreed to participate in this study and signed a written consent form; (2) subjects are healthy males and females aged 18-75. Exclusion criteria were: (1) nonsteroidal anti-inflammatory drugs (NSAIDs) and corticosteroids; (2) autoimmune or inflammatory related pathologies (cancer, diabetes, infections; (3) neurological or psychiatric disorder.

  • elaborate on how TBI severity was determined (e.g., Glasgow Coma Scale, imaging findings, duration of unconsciousness) and how patients were classified into the “moderate-to-severe” group.
    Response: Thank you for this comment. We have clarified in the Methods section how TBI severity was determined. Severity was classified according to established criteria based on acute clinical data. Specifically, we considered the initial Glasgow Coma Scale (GCS) score at hospital admission, duration of loss of consciousness and post-traumatic amnesia, Patients were classified as moderate-to-severe when they had a GCS ≤ 12 and/or pro-longed LOC (>30 minutes) or PTA (>24 hours). 

  • Regarding cytokine analysis:
    • Provide more detailed information on sample collection and storage procedures, including handling of serum samples, centrifugation protocols, and storage duration.
    • The phrase “serum was thawed upon performing experiments” raises concern about possible multiple freeze–thaw cycles, which may affect cytokine stability. Please clarify whether samples were aliquoted and frozen only once.
    • Time of sample storage is important, particularly for short half-life cytokines like IL-10. Please report the average or range of storage duration.
    • Note that blood is collected from participants, not serum. The correct terminology is that blood was drawn and processed to obtain serum for analysis.
      Response: Thank you for your suggestions: the paragraph regarding Serum Sample Collection has been modified providing more details on sample collection and storage procedures, clarifying time of sample storage. The terminology concerning blood collection has been corrected too as you suggested. The paragraph has been modified as follows:

      Line 450 Blood was drawn from all subjects enrolled in the study and processed to obtain serum for analysis. Serum was obtained from blood by centrifugation (2000 g × 10’ at room temperature) and stored within 1 hour and 30 minutes from the collection in several aliquots at -80 °C immediately after sampling in order to avoid freeze–thaw cycles and was thawed upon performing experiments.

  • The statistical analysis section should be expanded:
    • Specify which tests were used to assess normality (e.g., Shapiro-Wilk, Kolmogorov-Smirnov).
    • Indicate how normally and non-normally distributed variables were analyzed.
    • Clarify how correlation coefficients (e.g., Spearman’s rho) were interpreted in terms of strength and significance.
      Response: Thank you for your suggestion. Normality was assessed using Shapiro-Wilk test. Normally distributed data were analyzed using student’s t-test, whereas not-normally distributed data were analyzed by Mann–Whitney U test. Spearman's rank correlation strength is indicated by the magnitude of the correlation coefficient (ρ or rs), which ranges from -1 to 1. Values between 0.3 and 0.7 indicate a moderate positive relationship. Our results indicate a positive moderate correlation between IL-10 and sleep efficacy which is statistical significant (p-value = 0.04)

      The statistical analysis section was modified according to your comments as follows:

      Line 494. Quantitative data were assessed for normality by Shapiro-Wilk test. Normally distributed data were summarized as mean ± standard deviation (days from injury and age), whereas not-normally distributed data (Disability rating scale at admission, cytokines, neurotrophins, neurotransmitters) were summarized as median and interquartile range (IQR). Normally distributed data were analyzed using student’s t-test, whereas not-normally distributed data were analyzed by Mann–Whitney U test. Correlations were analyzed by Spearman’s rank correlation coefficient (Rsp).

      Data analysis was performed using the MedCalc statistical package (MedCalc Software bvba, Mariakerke, Belgium). p-values of less than 0.05 were considered statistically significant.

5. Conclusion
The study offers novel insights into peripheral biomarkers associated with PTCS following TBI, with potential implications for early diagnosis and monitoring. However, certain methodological details and clarifications are essential to enhance transparency and scientific rigor. I recommend major revisions before the manuscript can be considered for publication.

Reviewer 2 Report

Comments and Suggestions for Authors

The study addresses a relevant and current topic in clinical neuroscience and neurological rehabilitation, particularly by exploring biological markers associated with post-traumatic confusional state (PTCS) in patients with moderate to severe traumatic brain injury (TBI). The manuscript is of merit and has potential for publication, particularly due to its integrated approach involving inflammatory biomarkers, neuroendocrine function, neurotrophins, and sleep parameters. The laboratory methods and the use of actigraphy also add innovation and value to the study.

I would like to add some suggestions and considerations that may contribute to further improving the scientific rigor and clarity of the article. The main points are as follows:

- Limitations should be addressed in the manuscript: One is the sample size. A small sample size (29 patients with TBI (17 with PTCS and 12 without) and 34 controls) may compromise statistical power and generalizability of the findings. Therefore, this should be reported, and if possible, a sample size calculation should be established to indicate that the sample size is sufficient for inferences.

- It can be seen that there was no assessment of confounding variables, for example: Variables such as medication use (e.g., sedatives, corticosteroids), previous psychiatric/neurological comorbidities, and previously unidentified sleep disorders.

- More precise information about the ELISA is lacking: information about the lack of standardization; triplicate ELISA assays would be ideal; there is no information about standard curves or laboratory quality control.

- In the statistical tests, there is a lack of confidence intervals for the measurements; in some tests, there is no correction for multiple comparisons, which increases the risk of false positives (type I error).

- A regression analysis could be useful to control for the effects of age, sex, and time post-injury.

Author Response

Reviewer 2

The study addresses a relevant and current topic in clinical neuroscience and neurological rehabilitation, particularly by exploring biological markers associated with post-traumatic confusional state (PTCS) in patients with moderate to severe traumatic brain injury (TBI). The manuscript is of merit and has potential for publication, particularly due to its integrated approach involving inflammatory biomarkers, neuroendocrine function, neurotrophins, and sleep parameters. The laboratory methods and the use of actigraphy also add innovation and value to the study.

I would like to add some suggestions and considerations that may contribute to further improving the scientific rigor and clarity of the article. The main points are as follows:

  • Limitations should be addressed in the manuscript: One is the sample size. A small sample size (29 patients with TBI (17 with PTCS and 12 without) and 34 controls) may compromise statistical power and generalizability of the findings. Therefore, this should be reported, and if possible, a sample size calculation should be established to indicate that the sample size is sufficient for inferences.
    Response: Thank you for your kind suggestion. We are aware of the limit of sample size of TBI population; this point has been addressed in a paragraph in discussion section at the end of the manuscript as follows:

    Line 374 Limitations: this study has some limitations that must be acknowledged. First, the small sample size (29 patients with TBI (17 with PTCS and 12 without) and 34 controls) limits the statistical power of the analysis. Further, this study showed a sex imbalance distribution; finally, this study does not provide longitudinal follow-up to assess clinical outcomes or functional recovery. However, rehabilitation program on TBI patients is ongoing and data will be provided in a new work.

  • It can be seen that there was no assessment of confounding variables, for example: Variables such as medication use (e.g., sedatives, corticosteroids), previous psychiatric/neurological comorbidities, and previously unidentified sleep disorders.
    Response: Thank you for your comment. This point has been addressed in paragraph the exclusion criteria as follows:

    Line 399 Exclusion criteria included: (1) age <18 or >75 years; (2) a previous neurological or psychiatric disorder (particularly neurodegenerative or acquired conditions affecting cognitive domains, e.g., dementia); (3) autoimmune or inflammatory related pathologies (cancer, diabetes, infections); (4) medical instability; (5) the presence of a disorder of consciousness at IRU admission; (6) use of nonsteroidal anti-inflammatory drugs (NSAIDs) and corticosteroids; (7) previously diagnosed sleep disorders.

  • More precise information about the ELISA is lacking: information about the lack of standardization; triplicate ELISA assays would be ideal; there is no information about standard curves or laboratory quality control.
    Response: Thank you for your comment. All the experiments were performed in duplicate for standardization.  Standard curves for all the variables analyzed were linear.

    As it regards standardization, two samples with known concentrations were added in each plate and run in order to standardize the experiments.

  • In the statistical tests, there is a lack of confidence intervals for the measurements; in some tests, there is no correction for multiple comparisons, which increases the risk of false positives (type I error).
    Response: Thank you for your kind suggestions; I added the IQR in all results.

    As it regards the correction for multiple comparisons, unlikely, the number of the enrolled subject is not so big and for this reason some significativities disappear when correction is applied. So, I prefer to maintain the results without the correction but I modified the figure legends adding this information.

  • A regression analysis could be useful to control for the effects of age, sex, and time post-injury.
    Response: Thank you for your kind suggestion. We performed a regression analysis for age, DRS and time post injury. Results showed no effect of these variables on cytokine, neurotransmitters and neurotrophin concentrations.

Round 2

Reviewer 1 Report

Comments and Suggestions for Authors

The authors carefully revised the manuscript, taking into account the reviewers' comments and suggestions. In its current version, the manuscript is ready for publication.

Comments on the Quality of English Language

The English could be improved to more clearly express the research.

Author Response

Dear reviewer,

thank you for your suggestion, but we prefer to proceed without any further changes. We would need at least 10 working days to get approval from the administrative office, which would delay the article's publication. Thank you.